# Dynamical Analysis of Charged Dissipative Cylindrical Collapse in Energy-Momentum Squared Gravity †

Muhammad Zeeshan Gul and Muhammad Sharif *

Department of Mathematics, Quaid-e-Azam Campus, University of the Punjab, Lahore 54590, Pakistan; mzeeshangul.math@gmail.com
* Correspondence: msharif.math@pu.edu.pk
† This paper is an extended version from the proceeding paper: Muhammad Sharif and Muhammad Gul. Dynamics of Anisotropic Cylindrical Collapse in Energy-Momentum Squared Gravity. In Proceedings of the 1st Electronic Conference on Universe, online, 22–28 February 2021.

**Abstract:** This paper investigates the dynamics of charged cylindrical collapse with the dissipative matter configuration in $f(\mathcal{R}, \mathcal{T}_{\alpha\beta}\mathcal{T}^{\alpha\beta})$ theory. This newly formulated theory resolves the primordial singularity and provides feasible cosmological results in the early universe. Moreover, its implications occur in high curvature regime where the deviations of energy-momentum squared gravity from general relativity is confirmed. We establish dynamical and transport equations through the Misner–Sharp and M*ü*ler–Israel Stewart techniques, respectively. We then couple these equations to examine the impact of effective fluid parameters and correction terms on the collapsing phenomenon. A connection between the modified terms, matter parameters, and Weyl tensor is also developed. To obtain conformal flatness, we choose a particular model of this theory and assume that dust matter with zero charge leads to conformal flatness and homogenous energy density. We found that the modified terms, dissipative matter, and electromagnetic field reduce the collapsing phenomenon.

**Keywords:** $f(\mathcal{R}, \mathcal{T}_{\alpha\beta}\mathcal{T}^{\alpha\beta})$ gravity; conformal flatness; dynamics; collapsing phenomenon; electromagnetic field

## 1. Introduction

Cosmological findings indicate that the universe was created by the expansion of matter and energy. Cosmologists have explored many secrets of the universe and it appears that a large part of this mysterious universe includes stars, clusters, planets, and galaxies. The hidden aspects of these celestial objects have inspired many researchers to investigate their viable characteristics in the background of cosmology and astrophysics. Cosmology also helps to explain different captivating phenomena, such as the big-bang, stellar evolution, and gravitational collapse.

Stellar evolution determines the different evolutionary aspects of a star, such as birth, death, and even age. The stability of a star remains preserved if the outward stress of matter is balanced by the attractive pull. This stable state of a self-gravitating object is disturbed once the inner fuel is exhausted and there is not sufficient stress to counter-balance the gravitational force. Consequently, the star collapses and new remnants are obtained, which are dense objects that are assumed to be very compact because of the huge masses and small radii.

Gravitational collapse is viewed as a captivating and widely debated issue in astrophysics and cosmology. Chandrasekhar [1] is a pioneer in the investigation of a star's collapse, who found that a star preserves its stable state when the outward-directed pressure and inward gravitational force counter-balance each other. Oppenheimer and Snyder [2] analyzed the dynamics of dust collapse and concluded that a black hole is the final outcome of gravitational collapse.

Herrera and Santos [3] applied the Misner–Sharp strategy to investigate spherical collapse via dynamical and transport equations. Herrera et al. [4] examined anisotropic cylindrical collapse and explored the impact of fluid parameters on the rate of collapse. Di Prisco et al. [5] analyzed the heat dissipation of anisotropic spherical collapse through dynamical and transport equations. Tewari and Charan [6] examined spherical collapse in the presence of imperfect fluid distribution.

The current cosmic acceleration has been the most stunning and dazzling consequence for cosmologists over the past two decades. Scientists claim that this acceleration is caused by an ambiguous force dubbed as dark energy, which has repulsive effects. This mysterious force has motivated numerous researchers to investigate its unknown attributes. In this regard, modified theories are regarded as the most important proposals to reveal the mysterious universe.

Such proposals can be established by the implementation of scalar invariants and their corresponding functions in the geometric section of the Einstein–Hilbert action. The $f(\mathcal{R})$ theory is the simplest modification of general relativity (GR). A comprehensive literature [7–9] is available regarding the the physically realistic attributes of this modified theory. This can further be generalized through interactions among geometric and matter parts known as curvature-matter coupled theories.

Such interactions determine the rotation curves of galaxies as well as different cosmic stages. These theories are non-conserved, which provides the presence of an extra force. These coupling proposals are very useful to comprehend the cosmic acceleration and cryptic nature of dark components. Harko et al. [10] introduced such interactions in the $f(\mathcal{R})$ theory named as $f(\mathcal{R}, \mathcal{T})$ gravity. The non-minimal interaction of geometry and matter was developed in [11] known as $f(\mathcal{R}, \mathcal{T}, \mathcal{R}_{\alpha\beta} \mathcal{T}^{\alpha\beta})$ theory.

The presence of singularities in GR is a major problem because of its prediction at large scales, where GR is not valid due to the quantum effects. Nevertheless, quantum theory does not have a specific method. Accordingly, $f(\mathcal{R}, \mathbf{T}^2)$ theory (also dubbed as energy-momentum squared gravity (EMSG)) has been established by incorporating $\mathcal{T}^{\alpha\beta} \mathcal{T}_{\alpha\beta} = \mathbf{T}^2$ in the generic action [12].

This includes quadratic entities of the matter variables and their multiples in the equations of motion, which help to explain various fascinating cosmological results. This newly developed proposal has a bounce with finite maximal matter density and a minimal scale factor at early times. As a result, this can resolve the primordial singularity with a non-quantum description. It is noteworthy that this proposal resolves the spacetime singularities but that cosmological evolution remains unaffected.

Board and Barrow [13] studied the exact solutions of isotropic spacetime and investigated their behavior with cosmic growth, presence and absence of singularities. Nari and Roshan [14] explored the feasible and stable compact stars in this context. Bahamonde et al. [15] analyzed the minimal and non-minimal EMSG models and concluded that these models determine the cosmic acceleration. Barbar et al. [16] investigated physically viable conditions of the bouncing universe corresponding to a specific EMSG model.

Singh et al. [17] analyzed the geometry of compact objects in the existence of quark matter. Recently, we have discussed the Noether symmetry approach in this framework and investigated the viable behavior of cosmological solutions through different cosmological parameters. We have also studied the dynamics and stability of dense objects [18–22]. It is found that modified EMSG terms boost the system's stability and hence prevent the collapsing phenomenon. The above literature makes it clear that this theory needs more attention in different context.

The electromagnetic field is a key aspect in the evolution and stability of self-gravitating objects. The electromagnetic forces are major components to minimize the attractive force. Bekenstein [23] investigated the collapsing phenomenon of charged sphere collapse and found that electromagnetic field yields repulsive effects on the collapse rate. Esculpi and Aloma [24] analyzed spherical collapse in the presence of charged anisotropic matter con-

figuration and found that anisotropic factor as well as charge minimize the collapse rate. Sharif and Fatima [25] discussed the collapsing phenomenon of the anisotropic cylindrical star. The dynamics of planar symmetry with electromagnetic field has been investigated in [26].

The Weyl tensor plays a key role in the evolution of stellar structures, which determines the geometric curvature. We obtain a conformally flat universe when the Weyl tensor vanishes. Penrose [27] studied spherical collapse and developed a connection between matter variables and Weyl tensor. Herrera et al. [28] examined conformally flat spacetime with the help of a Weyl tensor. Sharif and Bhatti [29,30] explored dynamics of spherical spacetime and established a relation among anisotropy, electromagnetic field and Weyl tensor. The Weyl tensor can be formulated for spherical as well as cylindrical spacetimes with anisotropic matter distribution, which may affect the collapse rate [31,32].

The existence of cylindrical gravitational waves facilitates the cylindrical symmetries that yield important consequences about self-gravitating structures. The significance of cylindrical geometries has drawn the attention of many astronomers to examine different astrophysical phenomena. Bronnikov and Kovalchuk [33] were the pioneers in the study of cylindrical stellar structure. The collapsing phenomenon of cylindrical stars with distinct fluid configurations has also been widely discussed in alternative gravitational theories.

Shamir and Fayyaz [34] discussed the dynamics of dissipative cylindrical collapse, including charge anisotropic matter, and found that collapse was the key aspect to discuss the late time acceleration in $f(\mathcal{R})$ theory. Zubair et al. [35] analyzed the collapsing phenomenon of anisotropic cylindrical stars in $f(\mathcal{R}, \mathcal{T})$ theory. Khan et al. [36] studied the charged anisotropic cylindrical collapse in $f(\mathcal{R}, G)$ theory, where $G$ stands for the Gauss–Bonnet term. Sharif et al. [37–43] investigated the collapse of a cylindrical object with distinct fluid configurations in various modified theories and discussed the role of the matter parameters as well as the correction terms on the collapse rate.

This paper studies the charged dissipative cylindrical collapse in the background of $f(\mathcal{R}, \mathcal{T}_{\alpha\beta}\mathcal{T}^{\alpha\beta})$ gravity. The paper is organized as follows. We establish the corresponding equations of motion and matching conditions in Section 2. In Section 3, we use the Misner–Sharp formalism to study the dynamics of the system. In Section 4, we establish the transport equations and analyze the impact of correction terms on the rate of collapse. Section 5 is devoted to formulating a link between the modified terms, physical quantities, and Weyl tensor. We summarize our results in the last section.

## 2. $f(\mathcal{R}, \mathcal{T}_{\alpha\beta}\mathcal{T}^{\alpha\beta})$ Gravity and Junction Conditions

We establish the field equations and Darmois junction conditions with imperfect matter configuration in this section. In the presence of an electromagnetic field, this theory takes the following action [12]

$$\mathcal{I} = \int \left( \frac{f(\mathcal{R}, \mathcal{T}_{\alpha\beta}\mathcal{T}^{\alpha\beta})}{2\kappa^2} + \mathcal{L}_M + \mathcal{L}_E \right) \sqrt{-g} d^4 x \tag{1}$$

where the coupling constant, determinant of the line element, Lagrangian of matter, and electromagnetic field are denoted by $\kappa^2$, $g$, $\mathcal{L}_M$, and $\mathcal{L}_E$, respectively. Here, $\mathcal{L}_E = \frac{-1}{16\pi}\mathbb{F}^{\alpha\beta}\mathbb{F}_{\alpha\beta}$, $\mathbb{F}_{\alpha\beta} = \varphi_{\beta,\alpha} - \varphi_{\alpha,\beta}$ manifests the Maxwell field tensor, and the four-potential is represented by $\varphi_\alpha$. We assumed that some useful results will be obtained to study the stellar structures due to the matter-dominated era. The following equations of motion are obtained from the variation of action corresponding to the metric tensor.

$$\mathcal{R}_{\alpha\beta}f_\mathcal{R} + g_{\alpha\beta}\Box f_\mathcal{R} - \nabla_\alpha\nabla_\beta f_\mathcal{R} - \frac{1}{2}g_{\alpha\beta}f = \mathcal{T}_{\alpha\beta}^{(M)} + \mathcal{T}_{\alpha\beta}^{(E)} - \Theta_{\alpha\beta}f_{\mathcal{T}_{\alpha\beta}\mathcal{T}^{\alpha\beta}}, \tag{2}$$

where $\Box = \nabla_\mu \nabla^\mu$, $f \equiv f(\mathcal{R}, \mathcal{T}_{\alpha\beta}\mathcal{T}^{\alpha\beta})$, $f_{\mathcal{T}_{\alpha\beta}\mathcal{T}^{\alpha\beta}} = \frac{\partial f}{\partial \mathcal{T}_{\alpha\beta}\mathcal{T}^{\alpha\beta}}$, $f_{\mathcal{R}} = \frac{\partial f}{\partial \mathcal{R}}$,

$$\Theta_{\alpha\beta} = -2\mathcal{L}_m \left( \mathcal{T}_{\alpha\beta} - \frac{1}{2}g_{\alpha\beta}\mathcal{T} \right) - 4\frac{\partial^2 \mathcal{L}_M}{\partial g^{\alpha\beta}\partial g^{\mu\nu}}\mathcal{T}^{\mu\nu} - \mathcal{T}\mathcal{T}_{\alpha\beta} + 2\mathcal{T}_\alpha^\mu \mathcal{T}_{\beta\mu},$$

and $\mathcal{T}_{\alpha\beta}^{(E)}$ defines the energy-momentum tensor of electromagnetic field as

$$\mathcal{T}_{\alpha\beta}^{(E)} = \frac{1}{4\pi} \left( \mathbb{F}_\alpha^\mu \mathbb{F}_{\beta\mu} - \frac{1}{4}g_{\alpha\beta}\mathbb{F}^{\mu\nu}\mathbb{F}_{\mu\nu} \right).$$

To examine the stellar structures, we take into account cylindrical spacetime as

$$ds_-^2 = -\mathcal{W}^2(t,r)dt^2 + \mathcal{X}^2(t,r)dr^2 + \mathcal{Y}^2(t,r)d\phi^2 + dz^2. \tag{3}$$

The distribution of fluid is defined by the energy-momentum tensor and every non-zero element provides physically viable characteristics of the dynamical system. Here, we consider imperfect matter configurations as

$$\begin{aligned}
\mathcal{T}_{\alpha\beta}^{(M)} &= (\text{æ} + P_r)\mathcal{U}_\alpha\mathcal{U}_\beta - (P_r - P_{\text{Œ}})\mathcal{V}_\alpha\mathcal{V}_\beta - (P_r - P_z)\mathcal{S}_\alpha\mathcal{S}_\beta \\
&\quad - (g_{\alpha\beta} + \mathcal{U}_\alpha\mathcal{U}_\beta)\zeta\Theta + h_\alpha\mathcal{U}_\beta + h_\beta\mathcal{U}_\alpha + P_r g_{\alpha\beta},
\end{aligned} \tag{4}$$

where æ, $P_r$, $P_\phi$, and $P_z$ determine the energy density and principal pressures, $\zeta$ stands for the coefficient of bulk viscosity, $h_\alpha$ represents the heat flux, and $\Theta$ defines the expansion scalar. Here, $\mathcal{U}_\alpha$, $\mathcal{V}_\alpha$, $\mathcal{S}_\alpha$, $\Theta$, and $h_\alpha$ are determined as

$$\mathcal{U}_\alpha = -\mathcal{W}\delta_\alpha^0, \quad \mathcal{V}_\alpha = \mathcal{Y}\delta_\alpha^2, \quad \mathcal{S}_\alpha = \delta_\alpha^3, \quad \Theta = \mathcal{U}_{;\alpha}^\alpha, \quad h_\alpha = h\mathcal{X}\delta_\alpha^1,$$

which obey the following relations

$$\mathcal{U}_\alpha\mathcal{U}^\alpha = -1, \quad \mathcal{V}_\alpha\mathcal{V}^\alpha = 1 = \mathcal{S}_\alpha\mathcal{S}^\alpha, \quad \mathcal{U}_\alpha\mathcal{V}^\alpha = \mathcal{V}_\alpha\mathcal{S}^\alpha = \mathcal{U}_\alpha\mathcal{S}^\alpha = 0.$$

Rearranging Equation (2), we have

$$\mathcal{G}_{\alpha\beta} = \frac{1}{f_{\mathcal{R}}} \left( \mathcal{T}_{\alpha\beta}^{(M)} + T_{\alpha\beta}^{(E)} + \mathcal{T}_{\alpha\beta}^{(EMSG)} \right) = \mathcal{T}_{\alpha\beta}^{(eff)}, \tag{5}$$

where $\mathcal{G}_{\alpha\beta}$ is the Einstein tensor, and $\mathcal{T}_{\alpha\beta}^{(EMSG)}$ are the additional impacts of EMSG that include the higher-order curvature terms due to the modification in the curvature part, which are named as correction terms as

$$\mathcal{T}_{\alpha\beta}^{(EMSG)} = \frac{1}{2}g_{\alpha\beta}(f - \mathcal{R}f_{\mathcal{R}}) + (\nabla_\alpha\nabla_\beta - g_{\alpha\beta}\Box)f_{\mathcal{R}} - \Theta_{\alpha\beta}f_{\mathcal{T}_{\alpha\beta}\mathcal{T}^{\alpha\beta}}. \tag{6}$$

The Maxwell field equations are expressed as

$$\mathbb{F}_{[\alpha\beta;\nu]} = 0, \quad \mathbb{F}_{;\beta}^{\alpha\beta} = 4\pi\mathcal{J}^\alpha, \tag{7}$$

where $\mathcal{J}^\alpha = \xi\mathcal{U}^\alpha$ represents the four-current, and the charge density is defined by $\xi$. For the interior spacetime, the charge conservation law yields

$$\mathcal{Q}(r) = 4\pi \int_0^r \xi\mathcal{X}\mathcal{Y}^2 dr,$$

where $\mathcal{Q}(\mathrm{r})$ manifests the total charge in the cylindrical star. The electric field intensity is determined as

$$\mathbb{E} = \frac{\mathcal{Q}(\mathrm{r})}{4\pi \mathcal{Y}^2}. \tag{8}$$

As a result, the Maxwell field equations can be written as

$$\dot{\varphi}' - \left(\frac{\dot{\mathcal{W}}}{\mathcal{W}} + \frac{\dot{\mathcal{X}}}{\mathcal{X}} - \frac{\dot{\mathcal{Y}}}{\mathcal{Y}}\right)\varphi' = 0, \tag{9}$$

$$\varphi'' - \left(\frac{\mathcal{W}'}{\mathcal{W}} + \frac{\mathcal{X}'}{\mathcal{X}} - \frac{\mathcal{Y}'}{\mathcal{Y}}\right)\varphi' = 4\pi\xi\mathcal{W}\mathcal{X}^2, \tag{10}$$

where *prime* and *dot* demonstrate the derivatives corresponding to r and t, respectively. The corresponding field equations are

$$\frac{1}{f_{\mathcal{R}}}\left(\mathcal{T}_{00}^{(M)} + \mathcal{T}_{00}^{(E)} + \mathcal{T}_{00}^{(EMSG)}\right) = \frac{1}{f_{\mathcal{R}}}\left(\text{æ} + 2\pi\mathbb{E}^2 + \frac{\mathcal{T}_{00}^{(EMSG)}}{\mathcal{W}^2}\right)$$
$$= \frac{\mathcal{X}'\mathcal{Y}'}{\mathcal{X}^3\mathcal{Y}} - \frac{\mathcal{Y}''}{\mathcal{X}^2\mathcal{Y}} + \frac{\dot{\mathcal{X}}\dot{\mathcal{Y}}}{\mathcal{W}^2\mathcal{X}\mathcal{Y}}, \tag{11}$$

$$\frac{1}{f_{\mathcal{R}}}\left(\mathcal{T}_{11}^{(M)} + \mathcal{T}_{11}^{(E)} + \mathcal{T}_{11}^{(EMSG)}\right) = \frac{1}{f_{\mathcal{R}}}\left(\mathrm{P_r} - \zeta\Theta - 2\pi\mathbb{E}^2 + \frac{\mathcal{T}_{11}^{(EMSG)}}{\mathcal{X}^2}\right)$$
$$= \frac{\dot{\mathcal{W}}\dot{\mathcal{Y}}}{\mathcal{W}^3\mathcal{Y}} - \frac{\ddot{\mathcal{Y}}}{\mathcal{W}^2\mathcal{Y}} + \frac{\mathcal{W}'\mathcal{Y}'}{\mathcal{W}\mathcal{X}^2\mathcal{Y}}, \tag{12}$$

$$\frac{1}{f_{\mathcal{R}}}\left(\mathcal{T}_{22}^{(M)} + \mathcal{T}_{22}^{(E)} + \mathcal{T}_{22}^{(EMSG)}\right) = \frac{1}{f_{\mathcal{R}}}\left(\mathrm{P_\phi} - \zeta\Theta + 2\pi\mathbb{E}^2 + \frac{\mathcal{T}_{22}^{(EMSG)}}{\mathcal{Y}^2}\right)$$
$$= \frac{\dot{\mathcal{W}}\dot{\mathcal{X}}}{\mathcal{W}^3\mathcal{X}} + \frac{\mathcal{W}''}{\mathcal{W}\mathcal{X}^2} - \frac{\ddot{\mathcal{X}}}{\mathcal{W}^2\mathcal{X}} - \frac{\mathcal{W}'\mathcal{X}'}{\mathcal{W}\mathcal{X}^3}, \tag{13}$$

$$\frac{1}{f_{\mathcal{R}}}\left(\mathcal{T}_{33}^{(M)} + \mathcal{T}_{33}^{(E)} + \mathcal{T}_{33}^{(EMSG)}\right) = \frac{1}{f_{\mathcal{R}}}\left(\mathrm{P_z} - \zeta\Theta + 2\pi\mathbb{E}^2 + \mathcal{T}_{33}^{(EMSG)}\right)$$
$$= \frac{\dot{\mathcal{W}}\dot{\mathcal{Y}}}{\mathcal{W}^3\mathcal{Y}} - \frac{\ddot{\mathcal{X}}}{\mathcal{W}^2\mathcal{X}} - \frac{\ddot{\mathcal{Y}}}{\mathcal{W}^2\mathcal{Y}} + \frac{\dot{\mathcal{W}}\dot{\mathcal{X}}}{\mathcal{W}^3\mathcal{X}}$$
$$+ \frac{\mathcal{W}''}{\mathcal{W}\mathcal{X}^2} + \frac{\mathcal{Y}''}{\mathcal{X}^2\mathcal{Y}} + \frac{\mathcal{W}'\mathcal{Y}'}{\mathcal{W}\mathcal{Y}\mathcal{X}^2} - \frac{\mathcal{W}'\mathcal{X}'}{\mathcal{W}\mathcal{X}^3}$$
$$- \frac{\mathcal{Y}'\mathcal{X}'}{\mathcal{Y}\mathcal{X}^3} - \frac{\dot{\mathcal{X}}\dot{\mathcal{Y}}}{\mathcal{W}^2\mathcal{X}\mathcal{Y}}, \tag{14}$$

$$\frac{1}{f_{\mathcal{R}}}\left(h - \frac{\mathcal{T}_{01}^{(EMSG)}}{\mathcal{W}\mathcal{X}}\right) = \frac{\dot{\mathcal{Y}}'}{\mathcal{W}\mathcal{X}\mathcal{Y}} - \frac{\mathcal{Y}'\dot{\mathcal{X}}}{\mathcal{W}\mathcal{X}^2\mathcal{Y}} - \frac{\mathcal{W}'\dot{\mathcal{Y}}}{\mathcal{W}^2\mathcal{X}\mathcal{Y}}. \tag{15}$$

These equations describe how mass, fluid parameters, and gravity bend spacetime. The values of the correction terms $\mathcal{T}_{00}^{(EMSG)}$, $\mathcal{T}_{11}^{(EMSG)}$, $\mathcal{T}_{22}^{(EMSG)}$, $\mathcal{T}_{33}^{(EMSG)}$, and $\mathcal{T}_{01}^{(EMSG)}$ are given in Appendix A. The term $\left(\text{æ} + 2\pi\mathbb{E}^2 + \frac{\mathcal{T}_{00}^{(EMSG)}}{\mathcal{W}^2}\right)$ depicts the effective energy density while the factors $\left(\mathrm{P_r} - \zeta\Theta - 2\pi, \mathbb{E}^2 + \frac{\mathcal{T}_{11}^{(EMSG)}}{\mathcal{X}^2}\right)$, $\left(\mathrm{P_\phi} - \zeta\Theta + 2\pi\mathbb{E}^2 + \frac{\mathcal{T}_{22}^{(EMSG)}}{\mathcal{Y}^2}\right)$, $\left(\mathrm{P_z} - \zeta\Theta + 2\pi\mathbb{E}^2 + \mathcal{T}_{33}^{(EMSG)}\right)$, and $\left(h - \frac{\mathcal{T}_{01}^{(EMSG)}}{\mathcal{W}\mathcal{X}}\right)$ represent the effective principal pressures and the heat energy. The C-energy for the interior geometry is determined as [44,45]

$$\mathrm{E} = \mathfrak{m}(\mathrm{t}, \mathrm{r}) = \frac{1}{8}\left(1 - \mathbb{L}^{-2}\nabla^\beta \mathfrak{r}\nabla_\beta \mathfrak{r}\right), \tag{16}$$

where $\mathbb{L}^2 = \varsigma_{(3)\beta}\varsigma_{(3)}^{\beta}$, $\mathfrak{r} = v\mathbb{L}$, and $v^2 = \varsigma_{(2)\beta}\varsigma_{(2)}^{\beta}$ represent the specific length, circumference radius, and areal radius of cylindrical geometry. The entities $\varsigma_{(2)} = \frac{\partial}{\partial\phi}$ and $\varsigma_{(3)} = \frac{\partial}{\partial z}$ define the Killing vectors. Manipulating Equation (16), we have

$$\mathrm{E} = \frac{\mathbb{L}}{8}\left(1 + \frac{\dot{\mathcal{Y}}^2}{\mathcal{W}^2} - \frac{\mathcal{Y}'^2}{\mathcal{X}^2}\right) + \frac{\mathcal{Q}^2}{2\mathcal{Y}}. \tag{17}$$

The external spacetime is considered as

$$ds_+^2 = -\left(\frac{\mathfrak{q}^2(v)}{\mathrm{R}^2} - \frac{2\mathcal{M}(v)}{\mathrm{R}}\right)dv^2 - 2dvd\mathrm{R} + \mathrm{R}^2\left(d\phi^2 + \lambda^2 dz^2\right). \tag{18}$$

Here, $\mathcal{M}$, $\mathfrak{q}$, and R exhibit the mass, charge, and radius of the external geometry, whereas $\lambda$ is constant with dimensions of $L^{-1}$. For the smooth matching of the interior and exterior geometries, we consider Darmois junction conditions that yield

$$\mathcal{M} - \mathfrak{m} = \frac{\mathbb{L}}{8} \quad \Longleftrightarrow \quad \mathcal{Q} = \mathfrak{q}, \quad \mathbb{L} = 4R_{\Sigma}, \tag{19}$$

$$\mathrm{P_r} - \zeta\Theta - 2\pi\mathbb{E}^2 + \frac{\mathcal{T}_{11}^{(EMSG)}}{\mathcal{X}^2} = h - \frac{\mathcal{T}_{01}^{(EMSG)}}{\mathcal{W}\mathcal{X}} \quad \Longleftrightarrow \quad \mathcal{M} = 0. \tag{20}$$

These equations yield both the (*necessary/sufficient*) conditions for the smooth matching of both geometries. Equation (19) identifies that the masses of both spacetimes differ by $\frac{\mathbb{L}}{8}$, which is due to the least unsatisfactory definition of the Throne C-energy. Equation (20) determines the correlations among the effective radial pressure, heat flux, electromagnetic field, and the correction terms.

## 3. Dynamics of the Cylindrical Star

Here, we study the dynamics of the system via a dynamical equation defined as

$$\left(\mathcal{T}_{\alpha\beta}^{(\mathcal{M})} + \mathcal{T}_{\alpha\beta}^{(E)} + \mathcal{T}_{\alpha\beta}^{(EMSG)}\right)_{;\beta}h_\alpha = 0.$$

Manipulating this dynamical equation, we have

$$\frac{\mathcal{W}'}{\mathcal{X}^2\mathcal{W}}\left(\ae + \mathrm{P_r} - \zeta\Theta + \frac{\mathcal{T}_{00}^{(EMSG)}}{\mathcal{W}^2} + \frac{\mathcal{T}_{11}^{(EMSG)}}{\mathcal{X}^2}\right) - \frac{1}{\mathcal{W}\mathcal{X}}\left(\frac{\mathcal{T}_{01}^{(EMSG)}}{\mathcal{W}\mathcal{X}}\right)^{\cdot}$$

$$+ \frac{1}{\mathcal{X}^2}\left(\mathrm{P_r} - \zeta\Theta + \frac{\mathcal{T}_{11}^{(EMSG)}}{\mathcal{X}^2}\right)' + \frac{1}{\mathcal{W}\mathcal{X}}\left(h - \frac{\mathcal{T}_{01}^{(EMSG)}}{\mathcal{W}\mathcal{X}}\right)\frac{\dot{\mathcal{Y}}}{\mathcal{Y}} + \frac{h}{\mathcal{W}}$$

$$\frac{\mathcal{Y}'}{\mathcal{Y}\mathcal{X}^2}\left(\mathrm{P_r} - \mathrm{P_\phi} + \frac{\mathcal{T}_{11}^{(EMSG)}}{\mathcal{X}^2} - \frac{\mathcal{T}_{22}^{(EMSG)}}{\mathcal{Y}^2}\right) + \frac{\mathcal{Q}}{\mathcal{X}^2\mathcal{Y}^2}\left(\mathbb{E}'\mathcal{Y} - \mathcal{Y}'\mathbb{E}\right)$$

$$+ \frac{3\ddot{\mathcal{X}}h}{\mathcal{W}\mathcal{X}} = 0. \tag{21}$$

This equation is helpful to examine the variations in the evolution of stellar structures. In order to discuss the dynamics of the system, the proper derivatives corresponding to t and r are defined as [46,47]

$$\mathfrak{D}_t = \frac{1}{\mathcal{W}}\frac{\partial}{\partial\mathrm{t}}, \quad \mathfrak{D}_r = \frac{1}{\mathcal{Y}'}\frac{\partial}{\partial\mathrm{r}}. \tag{22}$$

The velocity of matter is determined as

$$\vartheta = \mathfrak{D}_t(\mathcal{Y}) = \frac{\dot{\mathcal{Y}}}{\mathcal{W}} < 0. \tag{23}$$

Using Equations (17) and (23), we have

$$\frac{\mathcal{Y}'}{\mathcal{X}} = \left(1 + \vartheta^2 - \frac{8\mathfrak{m}}{\mathbb{L}} + \frac{4\mathcal{Q}^2\mathbb{L}}{\mathcal{Y}}\right)^{\frac{1}{2}} = \psi.$$

The proper temporal derivative of C-energy turns out to be

$$
\begin{aligned}
\mathfrak{D}_t(E) &= -\frac{\mathcal{Y}\mathbb{L}}{4f_\mathcal{R}}\left\{\left(P_r - \zeta\Theta - 2\pi\mathbb{E}^2 + \frac{\mathcal{T}_{11}^{(EMSG)}}{\mathcal{X}^2}\right)\vartheta + \psi(h \right. \\
&\quad \left. - \frac{\mathcal{T}_{01}^{(EMSG)}}{\mathcal{W}\mathcal{X}}\right)\right\} - \vartheta\frac{\mathcal{Q}^2}{2\mathcal{Y}^2}.
\end{aligned}
$$

This provides the variation of the total energy in the cylindrical star. This relation manifests the influence of the modified terms, the electromagnetic field, heat flux, and effective radial stress on the collapsing phenomenon. The first factor in the round bracket becomes positive due to the presence of ($\vartheta$), which enhances the total energy of the system because of the outward effective pressure. The next entity in the round bracket implies that heat dissipates from the system, and hence the total energy in the collapsing source reduces. The last entity exhibits the Coulomb repulsive effect that diminishes the entire energy in the cylindrical star.

Next, we discuss how the total energy varies between the adjoining cylindrical surfaces. The rate of change of C-energy corresponding to the radial coordinate becomes

$$
\begin{aligned}
\mathfrak{D}_r(E) &= \frac{\mathcal{Y}\mathbb{L}}{4f_\mathcal{R}}\left\{\left(\ae + 2\pi\mathbb{E}^2 + \frac{\mathcal{T}_{00}^{(EMSG)}}{\mathcal{W}^2}\right) + \left(h - \frac{\mathcal{T}_{01}^{(EMSG)}}{\mathcal{W}\mathcal{X}}\right)\frac{\vartheta}{\psi}\right\} \\
&\quad + \frac{\mathcal{Q}}{\mathcal{Y}}D_r(\mathcal{Q}) - \frac{\mathcal{Q}^2}{2\mathcal{Y}^2}.
\end{aligned}
$$

The first factor in the round bracket contributes to the impact of the effective energy density on the collapse rate. The total energy of the system increases due to the presence of the energy density, which plays a role in the work performed. The next entity manifests the existence of heat energy, which dissipates from the system due to the velocity of the fluid parameters. The last factor shows that the total energy between the adjoining cylindrical surfaces reduces due to the Coulomb repulsive effects.

The collapsing source's acceleration is defined as

$$
\begin{aligned}
\mathfrak{D}_t(\vartheta) &= -\frac{\mathcal{Y}}{f_\mathcal{R}}\left(P_r - \zeta\Theta - 2\pi\mathbb{E}^2 + \frac{\mathcal{T}_{11}^{(EMSG)}}{\mathcal{X}^2}\right) + \frac{\psi}{\mathcal{X}}\frac{\mathcal{W}'}{\mathcal{W}} + \frac{\mathcal{Q}^2}{2\mathcal{Y}^3} \\
&\quad - \frac{\mathfrak{m}}{\mathcal{Y}^2} + \frac{\mathbb{L}}{8\mathcal{Y}^2}\left(1 + \vartheta^2 - \psi^2\right). \tag{24}
\end{aligned}
$$

Using Equation (21), we have

$$
\begin{aligned}
\frac{\mathcal{W}'}{\mathcal{W}} &= \left( \text{æ} + P_r - \zeta\Theta + \frac{\mathcal{T}_{00}^{(EMSG)}}{\mathcal{W}^2} + \frac{\mathcal{T}_{11}^{(EMSG)}}{\mathcal{X}^2} \right)^{-1} \left\{ \frac{\mathcal{X}}{\mathcal{W}} \left( \frac{\mathcal{T}_{01}^{(EMSG)}}{\mathcal{W}\mathcal{X}} \right)^{\cdot} \right. \\
&+ \frac{\mathcal{X}}{\mathcal{W}} \left( h - \frac{\mathcal{T}_{01}^{(EMSG)}}{\mathcal{W}\mathcal{X}} \right) \frac{\ddot{\mathcal{Y}}}{\mathcal{Y}} - \left( P_r - \zeta\Theta + \frac{\mathcal{T}_{11}^{(EMSG)}}{\mathcal{X}^2} \right)' - \frac{\dot{h}\mathcal{X}^2}{\mathcal{W}} \\
&- \frac{\mathcal{Y}'}{\mathcal{Y}} \left( P_r - P_\phi + \frac{\mathcal{T}_{11}^{(EMSG)}}{\mathcal{X}^2} - \frac{\mathcal{T}_{22}^{(EMSG)}}{\mathcal{Y}^2} \right) - \frac{\mathcal{Q}}{\mathcal{Y}^2} \left( \mathbb{E}'\mathcal{Y} - \mathcal{Y}'\mathbb{E} \right) \\
&- \left. \frac{3\mathcal{X}\ddot{\mathcal{X}}h}{\mathcal{W}} - \right\}.
\end{aligned}
$$

Substituting this value in Equation (24), we obtain

$$
\begin{aligned}
\mathfrak{D}_t(\vartheta) &\left( \text{æ} + P_r - \zeta\Theta + \frac{\mathcal{T}_{00}^{(EMSG)}}{\mathcal{W}^2} + \frac{\mathcal{T}_{11}^{(EMSG)}}{\mathcal{X}^2} \right) = -\left( \text{æ} + P_r - \zeta\Theta \right. \\
&+ \frac{\mathcal{T}_{00}^{(EMSG)}}{\mathcal{W}^2} + \frac{\mathcal{T}_{11}^{(EMSG)}}{\mathcal{X}^2} \right) \left\{ \frac{\mathfrak{m}}{\mathcal{Y}^2} + \frac{\mathcal{Y}}{f_\mathcal{R}} \left( P_r - \zeta\Theta - 2\pi\mathbb{E}^2 + \frac{\mathcal{T}_{11}^{(EMSG)}}{\mathcal{X}^2} \right) \right. \\
&- \frac{\mathcal{Q}^2}{2\mathcal{Y}^3} - \frac{\mathbb{L}}{8\mathcal{Y}^2} - \frac{\mathbb{L}\vartheta^2}{8\mathcal{Y}^2} \right\} - \psi^2 \left\{ \left( P_r - P_\phi + \frac{\mathcal{T}_{11}^{(EMSG)}}{\mathcal{X}^2} + \frac{\mathcal{T}_{22}^{(EMSG)}}{\mathcal{Y}^2} \right) \right. \\
&- \frac{\mathcal{Q}^2}{2\mathcal{Y}^3} - \frac{\mathbb{L}}{8\mathcal{Y}^2} \left( \text{æ} + P_r - \zeta\Theta + \frac{\mathcal{T}_{00}^{(EMSG)}}{\mathcal{W}^2} + \frac{\mathcal{T}_{11}^{(EMSG)}}{\mathcal{X}^2} \right) \left. \right\} - \psi \left\{ \left( P_r \right. \right. \\
&- \zeta\Theta + \frac{\mathcal{T}_{11}^{(EMSG)}}{\mathcal{X}^2} \right)' + \frac{3h\ddot{\mathcal{X}}}{\mathcal{W}} + \frac{\dot{\mathcal{Y}}}{\mathcal{Y}\mathcal{X}} \left( h - \frac{\mathcal{T}_{01}^{(EMSG)}}{\mathcal{W}\mathcal{X}} \right) + \frac{\mathcal{Q}\mathcal{Q}'}{2\pi\mathcal{X}\mathcal{Y}^2} \\
&+ \mathcal{X}\mathcal{D}_t \left( h - \frac{\mathcal{T}_{01}^{(EMSG)}}{\mathcal{W}\mathcal{X}} \right) \left. \right\}.
\end{aligned}
\tag{25}
$$

In the above Equation (25), the entity in the round bracket on the left side determines the inertial mass density of the collapsing source, whereas, on the right side, the first round bracket determines the gravitational mass density of the system. Hence, the equivalence principle is satisfied because of the equivalence in the inertial and gravitational masses. The first curly bracket includes the impact of the physical quantities and electromagnetic field on the collapsing phenomenon. The next one determines the role of the gravitational mass density, electric field intensity, and effective pressure in r and $\phi$ directions on the collapse rate. In the last bracket, the factor $\left( P_r - \zeta\Theta + \frac{\mathcal{T}_{11}^{(EMSG)}}{\mathcal{X}^2} \right)'$ manifests the impact of the effective gradient pressure and expansion scalar. All the remaining entities in this bracket describe the hydrodynamics of the system due to the energy dissipation and electromagnetic field.

## 4. Transport Equations

Transport equations provide information on how mass, heat, and momentum are evaluated during the collapsing phenomenon. The transport equation is defined as

$$
\varrho s^{\alpha\beta} \mathcal{U}^\gamma \bar{h}_{\beta;\gamma} = -\eta s^{\alpha\beta} \left( \tau_{,\beta} + \tau a_\beta \right) - \frac{1}{2}\eta\tau^2 \left( \frac{\varrho\mathcal{U}^\beta}{\eta\tau^2} \right)_{;\beta} \bar{h}^\alpha,
\tag{26}
$$

where $\bar{h} = \left( h - \frac{\mathcal{T}_{01}^{(EMSG)}}{\mathcal{W}\mathcal{X}} \right)$ determines the effective heat flux, $s^{\alpha\beta} = g^{\alpha\beta} + \mathcal{U}^{\alpha}\mathcal{U}^{\beta}$ is the projection tensor, and $\eta$, $\tau$, $\varrho$, and $a_{\beta}$ represent the thermal conductivity, temperature, relaxation time, and acceleration, respectively. Manipulating Equation (26), we obtain

$$
\begin{aligned}
\mathcal{X}\mathcal{D}_t\bar{h} &= -\frac{\eta\mathcal{X}\tau'}{\varrho} - \frac{\eta\mathcal{X}\tau}{\varrho}\left(\frac{\mathcal{W}'}{\mathcal{W}}\right) - \frac{\eta\tau^2\mathcal{X}^3}{2\varrho}\left(\frac{\varrho}{\eta\tau^2}\right)^{\cdot}\bar{h} + \frac{\dot{\mathcal{X}}}{\mathcal{W}}\bar{h} \\
&\quad - \frac{\mathcal{X}\bar{h}}{\mathcal{W}\varrho} - \frac{3\ddot{\mathcal{X}}\mathcal{X}\bar{h}}{2\mathcal{W}}.
\end{aligned}
\tag{27}
$$

Inserting the value of $\left(\frac{\mathcal{W}'}{\mathcal{W}}\right)$ in the above equation, we have

$$
\begin{aligned}
\mathcal{X}\mathcal{D}_t\left( h - \frac{\mathcal{T}_{01}^{(EMSG)}}{\mathcal{W}\mathcal{X}} \right) &= -\frac{\eta\mathcal{X}\tau'}{\varrho} - \frac{\eta\mathcal{X}^2\tau}{\varrho\psi}\mathcal{D}_t\mathcal{V} - \frac{\eta\tau^2\mathcal{X}^3}{2\varrho}\left(\frac{\varrho}{\eta\tau^2}\right)^{\cdot} \\
&\quad \times \left( h - \frac{\mathcal{T}_{01}^{EMSG}}{\mathcal{W}\mathcal{X}} \right) - \frac{\eta\tau\mathcal{X}^2}{\varrho\psi}\left\{ \frac{\mathrm{m}}{\mathcal{Y}^2} - \frac{\mathcal{Q}^2}{2\mathcal{Y}^3} \right. \\
&\quad + \frac{\mathcal{Y}}{f_{\mathcal{R}}}\left( \mathrm{P_r} - \zeta\Theta - 2\pi\mathbb{E}^2 + \frac{\mathcal{T}_{11}^{(EMSG)}}{\mathcal{X}^2} \right) \\
&\quad \left. - \frac{\mathbb{L}}{8}(\vartheta^2 + 1) \right\} + \left( \frac{\dot{\mathcal{X}}}{\mathcal{W}} - \frac{\mathcal{X}}{\tau\mathcal{W}} - \frac{3\mathcal{X}\ddot{\mathcal{X}}}{2\mathcal{W}} \right) \\
&\quad \times \left( h - \frac{\mathcal{T}_{01}^{EMSG}}{\mathcal{W}\mathcal{X}} \right) - \psi\frac{\eta\tau\mathcal{X}^2}{8\varrho\mathcal{Y}^2}.
\end{aligned}
$$

This equation describes how heat energy is evaluated with the passage of time. It also contributes the impact of the thermal conductivity, temperature, relaxation time, and gravitational force on the self-gravitating objects.

Manipulating Equation (27), we obtain

$$
\begin{aligned}
&\mathfrak{D}_t(\vartheta)\left( \mathbf{æ} + \mathrm{P_r} - \zeta\Theta + \frac{\mathcal{T}_{00}^{(EMSG)}}{\mathcal{W}^2} + \frac{\mathcal{T}_{11}^{(EMSG)}}{\mathcal{X}^2} - \frac{\eta\mathcal{X}^2\tau}{\varrho} \right) \\
&= -\left( \mathbf{æ} + \mathrm{P_r} - \zeta\Theta + \frac{\mathcal{T}_{00}^{(EMSG)}}{\mathcal{W}^2} + \frac{\mathcal{T}_{11}^{(EMSG)}}{\mathcal{X}^2} \right)\left\{ \frac{\mathrm{m}}{\mathcal{Y}^2} - \frac{\mathcal{Q}^2}{2\mathcal{Y}^3} - \frac{\mathbb{L}}{8\mathcal{Y}^2}\left( \vartheta^2 + 1 \right) \right. \\
&\quad \left. + \frac{\mathcal{Y}}{f_{\mathcal{R}}}\left( \mathrm{P_r} - \zeta\Theta - 2\pi\mathbb{E}^2 + \frac{\mathcal{T}_{11}^{(EMSG)}}{\mathcal{X}^2} \right) \right\}\left\{ 1 - \frac{\eta\tau\mathcal{X}^2}{\varrho}(\mathbf{æ} + \mathrm{P_r} - \zeta\Theta \right. \\
&\quad \left. + \frac{\mathcal{T}_{00}^{(EMSG)}}{\mathcal{W}^2} + \frac{\mathcal{T}_{11}^{(EMSG)}}{\mathcal{X}^2} \right)^{-1} \right\} - \psi^2\left\{ \left( \mathrm{P_r} - \mathrm{P_{\phi}} + \frac{\mathcal{T}_{11}^{(EMSG)}}{\mathcal{X}^2} + \frac{\mathcal{T}_{22}^{(EMSG)}}{\mathcal{Y}^2} \right) \right. \\
&\quad \left. - \frac{\mathcal{Q}^2}{2\mathcal{Y}^3} - \frac{\eta\tau\mathcal{X}^2\mathbb{L}}{8\varrho\mathcal{Y}^2} - \frac{\mathbb{L}}{8\mathcal{Y}^2}\left( \rho + \mathrm{P_r} - \zeta\Theta - 2\pi\mathbb{E}^2 + \frac{\mathcal{T}_{00}^{(EMSG)}}{\mathcal{W}^2} + \frac{\mathcal{T}_{11}^{(EMSG)}}{\mathcal{X}^2} \right) \right\} \\
&\quad - \psi\left\{ \frac{3h\ddot{\mathcal{X}}}{\mathcal{W}} - \frac{\eta\mathcal{X}\tau'}{\varrho} + \left( \mathrm{P_r} - \zeta\Theta + \frac{\mathcal{T}_{11}^{(EMSG)}}{\mathcal{X}^2} \right)' - \frac{\eta\tau^2\mathcal{X}^3}{2\varrho}\left( \frac{\varrho}{\eta\tau^2} \right)^{\cdot} \right. \\
&\quad \left. \times \left( h - \frac{\mathcal{T}_{01}^{EMSG}}{\mathcal{W}\mathcal{X}} \right) \right\} + \left( \frac{\dot{\mathcal{X}}}{\mathcal{W}} - \frac{\mathcal{X}}{\varrho\mathcal{W}} - \frac{3\mathcal{X}\ddot{\mathcal{X}}}{2\mathcal{W}} \right)\left( h - \frac{\mathcal{T}_{01}^{(EMSG)}}{\mathcal{W}\mathcal{X}} \right).
\end{aligned}
$$

This can be rearranged as

$$\mathfrak{D}_t(\vartheta)\left(\text{æ} + P_r - \zeta\Theta + \frac{\mathcal{T}_{00}^{(EMSG)}}{\mathcal{W}^2} + \frac{\mathcal{T}_{11}^{(EMSG)}}{\mathcal{X}^2}\right)(1 - Y) =$$

$$-\mathfrak{f}_{grav}(1 - Y) + \mathfrak{f}_{hyd} - \psi\left\{\left(P_r - \zeta\Theta + \frac{\mathcal{T}_{11}^{(EMSG)}}{\mathcal{X}^2}\right)' + \frac{3h\ddot{\mathcal{X}}}{\mathcal{W}}\right.$$

$$\left. - \frac{\eta\mathcal{X}\tau'}{\varrho} - \frac{\eta\tau^2\mathcal{X}^3}{2\varrho}\left(\frac{\varrho}{\eta\tau^2}\right)^{\cdot}\left(h - \frac{\mathcal{T}_{01}^{(EMSG)}}{\mathcal{W}\mathcal{X}}\right)\right\} + \left(h - \frac{\mathcal{T}_{01}^{(EMSG)}}{\mathcal{W}\mathcal{X}}\right)$$

$$\left(\frac{\dot{\mathcal{X}}}{\mathcal{W}} - \frac{\mathcal{X}}{\varrho\mathcal{W}} - \frac{3\mathcal{X}\ddot{\mathcal{X}}}{2\mathcal{W}}\right), \tag{28}$$

where

$$Y = \frac{\eta\tau\mathcal{X}^2}{\varrho}\left(\text{æ} + P_r - \zeta\Theta + \frac{\mathcal{T}_{00}^{(EMSG)}}{\mathcal{W}^2} + \frac{\mathcal{T}_{11}^{(EMSG)}}{\mathcal{X}^2}\right)^{-1},$$

$$\mathfrak{f}_{grav} = \left(\text{æ} + P_r - \zeta\Theta + \frac{\mathcal{T}_{00}^{(EMSG)}}{\mathcal{W}^2} + \frac{\mathcal{T}_{11}^{(EMSG)}}{\mathcal{X}^2}\right)\left\{\frac{\text{m}}{\mathcal{Y}^2} - \frac{\mathcal{Q}^2}{2\mathcal{Y}^3}\right.$$

$$\left. - \frac{\mathbb{L}}{8\mathcal{Y}^2}\left(\vartheta^2 + 1\right) + \frac{\mathcal{Y}}{f_{\mathcal{R}}}\left(P_r - \zeta\Theta - 2\pi\mathbb{E}^2 + \frac{\mathcal{T}_{11}^{(EMSG)}}{\mathcal{X}^2}\right)\right\},$$

$$\mathfrak{f}_{hyd} = -\psi^2\left\{\left(P_r - P_\phi + \frac{\mathcal{T}_{11}^{(EMSG)}}{\mathcal{X}^2} + \frac{\mathcal{T}_{22}^{(EMSG)}}{\mathcal{Y}^2}\right) - \frac{\mathcal{Q}^2}{2\mathcal{Y}^3} - \frac{\eta\tau\mathcal{X}^2\mathbb{L}}{8\varrho\mathcal{Y}^2}\right.$$

$$\left. - \frac{\mathbb{L}}{8\mathcal{Y}^2}\left(\rho + P_r - \zeta\Theta - 2\pi\mathbb{E}^2 + \frac{\mathcal{T}_{00}^{(EMSG)}}{\mathcal{W}^2} + \frac{\mathcal{T}_{11}^{(EMSG)}}{\mathcal{X}^2}\right)\right\}.$$

Equation (28) demonstrates the effects of various forces, including Newtonian ($\mathfrak{f}_{newtn}$), hydrodynamical ($\mathfrak{f}_{hyd}$), and gravitational ($\mathfrak{f}_{grav}$) forces, on the collapse rate. It is understood that energy dissipates in the form of radiation, convection, and conduction from the higher to lower energy state of the object. If photons gain energy from the higher phase of a star, then energy dissipates through radiation. The heat is dissipated by convection when photons do not hold all of the energy. In this phenomenon, hot gases travel towards the upper zone to radiate energy and cooler gases travel into the hot zone to attain energy. Every atom transfers its energy to the nearest atom because of continuous collisions, and energy dissipates through conduction.

Equation (28) determines how energy is dissipated by the hydrodynamical variables. This also includes an entity $(1 - Y)$ that justifies the principle of equivalence, while the term $(Y)$ is inversely proportional to the gravitational mass density. This relation manifests that the gravitational force is strongly affected by the term $(1 - Y)$, which leads to the following cases.

- If $Y < 1$ then $(1 - Y)$ becomes positive but, due to the minus sign, the action of the gravitational force may change and represents the behavior of the repulsive force that diminishes the collapse rate.
- If $Y > 1$ then $(1 - Y)$ becomes negative, which increases the collapsing phenomenon.
- If $Y = 1$ then the left side of Equation (28) and the gravitational force vanish, leading to

$$
\begin{aligned}
-\psi^2 &\left\{ \left( \mathrm{P_r} - \mathrm{P}_\phi + \frac{T_{11}^{(EMSG)}}{\mathcal{X}^2} + \frac{T_{22}^{(EMSG)}}{\mathcal{Y}^2} \right) - \frac{\mathcal{Q}^2}{2\mathcal{Y}^3} - \frac{\eta\tau\mathcal{X}^2\mathbb{L}}{8\varrho\mathcal{Y}^2} - \frac{\mathbb{L}}{8\mathcal{Y}^2} \right. \\
&\left. \times \left( \rho + \mathrm{P_r} - \zeta\Theta - 2\pi\mathbb{E}^2 + \frac{T_{00}^{(EMSG)}}{\mathcal{W}^2} + \frac{T_{11}^{(EMSG)}}{\mathcal{X}^2} \right) \right\} = \psi\left\{ \frac{3h\ddot{\mathcal{X}}}{\mathcal{W}} \right. \\
&\left. + \left( \mathrm{P_r} - \zeta\Theta + \frac{T_{11}^{(EMSG)}}{\mathcal{X}^2} \right)' - \frac{\eta\mathcal{X}\tau'}{\varrho} - \frac{\eta\tau^2\mathcal{X}^3}{2\varrho}\left(\frac{\varrho}{k\tau^2}\right)^{\cdot}\left( h - \frac{T_{01}^{(EMSG)}}{\mathcal{W}\mathcal{X}} \right) \right\} \\
&- \left( h - \frac{T_{01}^{(EMSG)}}{\mathcal{W}\mathcal{X}} \right)\left( \frac{\dot{\mathcal{X}}}{\mathcal{W}} - \frac{\mathcal{X}}{\varrho\mathcal{W}} - \frac{3\mathcal{X}\dot{\mathcal{X}}}{2\mathcal{W}} \right).
\end{aligned}
$$

The right side determines that energy of the system dissipates due to the inclusion of temperature and thermal conductivity. This also includes the effects of the correction terms and the bulk viscosity on the collapsing phenomenon. The left side is the hydrodynamical force, which supports the equilibrium state of the collapsing objects and, hence, reduces the collapse rate.

Now, we examine the impact of the modified terms and the factor $(Y)$ on the collapsing phenomenon in the presence of physically viable matter.

- If the correction terms are positive then the $(Y)$ will be less when compared to GR indicating that the term $(1 - Y)$ and the gravitational force are enhanced. Hence, the factor $\mathfrak{f}_{grav}(1 - Y)$ decreases due to the minus sign, which may diminish the collapse rate.
- If the correction terms have opposite signs, then we cannot find whether the collapsing processes increases or decreases.
- If the correction terms are negative then there are no physical effects.

## 5. Conformally Flat Spacetime

Here, we develop a relation between the fluid parameters, correction terms, and the Weyl scalar. The Weyl scalar $(\mathbb{C}^2 = \mathcal{C}_{\alpha\beta\mu\nu}\mathcal{C}^{\alpha\beta\mu\nu})$ in terms of Kretchmann, $(\mathfrak{R} = \mathcal{R}_{\alpha\beta\mu\nu}\mathcal{R}^{\alpha\beta\mu\nu})$, Ricci, and the scalar invariants is defined as [5]

$$
\mathbb{C}^2 = \frac{1}{3}\mathcal{R}^2 + \mathfrak{R} - 2\mathcal{R}^{\alpha\beta}\mathcal{R}_{\alpha\beta}. \tag{29}
$$

Manipulating $\mathfrak{R}$, we have

$$
\mathfrak{R} = 4\left\{ \frac{(R^{0101})^2}{W^4X^4} + \frac{(R^{0202})^2}{W^4Y^4} + \frac{(R^{1212})^2}{X^4Y^4} - \frac{(R^{1202})^2}{2W^2X^2Y^4} \right\}. \tag{30}
$$

For the cylindrical system, the values of Riemann tensor, Ricci tensor, and Ricci scalar in terms of the Einstein tensor are given as

$$
\begin{aligned}
\mathcal{R}^{0101} &= \frac{\mathcal{G}_{22}}{(\mathcal{W}\mathcal{X}\mathcal{Y})^2}, \quad \mathcal{R}^{0202} = \frac{\mathcal{G}_{11}}{(\mathcal{W}\mathcal{X}\mathcal{Y})^2}, \quad \mathcal{R}^{0212} = \frac{\mathcal{G}_{01}}{(\mathcal{W}\mathcal{X}\mathcal{Y})^2}, \\
\mathcal{R}^{1212} &= \frac{\mathcal{G}_{00}}{(\mathcal{W}\mathcal{X}\mathcal{Y})^2}, \quad \mathcal{R}_{00} = \mathcal{W}^2\left( \frac{\mathcal{G}_{11}}{\mathcal{X}^2} + \frac{\mathcal{G}_{22}}{\mathcal{Y}^2} \right), \quad \mathcal{R}_{01} = \mathcal{G}_{01}, \\
\mathcal{R}_{11} &= \mathcal{X}^2\left( \frac{\mathcal{G}_{00}}{\mathcal{W}^2} - \frac{\mathcal{G}_{22}}{\mathcal{Y}^2} \right), \quad \mathcal{R}_{22} = \mathcal{Y}^2\left( \frac{\mathcal{G}_{00}}{\mathcal{W}^2} - \frac{\mathcal{G}_{11}}{\mathcal{X}^2} \right), \\
\mathcal{R} &= 2\left( \frac{\mathcal{G}_{00}}{\mathcal{W}^2} - \frac{\mathcal{G}_{11}}{\mathcal{X}^2} - \frac{\mathcal{G}_{22}}{\mathcal{Y}^2} \right).
\end{aligned}
$$

Inserting the values of Reimann tensors in Equation (30), we have

$$\mathfrak{R} = 4\left(\frac{\mathcal{G}_{00}^2}{\mathcal{W}^4} + \frac{\mathcal{G}_{11}^2}{\mathcal{X}^4} + \frac{\mathcal{G}_{22}^2}{\mathcal{Y}^4} - \frac{4\mathcal{G}_{01}^2}{\mathcal{W}^2\mathcal{X}^2}\right).$$

Using the preceding equations, the Weyl scalar turns out to be

$$\mathbb{C}^2 = \frac{4}{3}\left(\frac{\mathcal{G}_{00}^2}{\mathcal{W}^4} + \frac{\mathcal{G}_{11}^2}{\mathcal{X}^4} + \frac{\mathcal{G}_{22}^2}{\mathcal{Y}^4} + \frac{\mathcal{G}_{00}\mathcal{G}_{11}}{\mathcal{W}^2\mathcal{X}^2} + \frac{\mathcal{G}_{00}\mathcal{G}_{22}}{\mathcal{W}^2\mathcal{Y}^2} - \frac{\mathcal{G}_{11}\mathcal{G}_{22}}{\mathcal{X}^2\mathcal{Y}^2}\right). \tag{31}$$

Using Equations (11)–(13), we obtain

$$
\begin{aligned}
\frac{\mathbb{C}\sqrt{3}}{2} =\ & \left[\left\{\frac{1}{f_\mathcal{R}}\left(\text{æ} + \mathrm{P_r} - \mathrm{P}_\phi - 2\pi\mathbb{E}^2 + \frac{\mathcal{T}_{00}^{(EMSG)}}{\mathcal{W}^2} + \frac{\mathcal{T}_{11}^{(EMSG)}}{\mathcal{X}^2} - \frac{\mathcal{T}_{22}^{(EMSG)}}{\mathcal{C}^2}\right)\right\}^2 \right. \\
& - \frac{1}{f_\mathcal{R}}\left\{\left(\mathrm{P_r} - 3\mathrm{P}_\phi + 2\zeta\Theta - 4\pi\mathbb{E}^2 + \frac{\mathcal{T}_{11}^{(EMSG)}}{\mathcal{X}^2} - \frac{3\mathcal{T}_{22}^{(EMSG)}}{\mathcal{C}^2}\right)\right. \\
& \times \left(\text{æ} + 2\pi\mathbb{E}^2 + \frac{\mathcal{T}_{00}^{(EMSG)}}{\mathcal{W}^2}\right) + \left(\mathrm{P_r} - \zeta\Theta - 2\pi\mathbb{E}^2 + \frac{\mathcal{T}_{11}^{(EMSG)}}{\mathcal{X}^2}\right) \\
& \left.\left. \times \left(\mathrm{P}_\phi - \zeta\Theta + 2\pi\mathbb{E}^2 + \frac{\mathcal{T}_{22}^{(EMSG)}}{\mathcal{Y}^2}\right)\right\}\right]^{\frac{1}{2}}.
\end{aligned}
\tag{32}
$$

The gravitational collapse in different modified theories has been analyzed in [48]. They determined that conformally flat spacetime provides homogeneous energy density and vice-versa. To analyze the viability of this result in $f(\mathcal{R}, \mathbf{T}^2)$ theory, we consider a particular type of function that gives the minimal coupling between the curvature and the matter parts defined as $f(\mathcal{R}, \mathbf{T}^2) = f_1(\mathcal{R}) + f_2(\mathbf{T}^2)$. When $f_2(\mathbf{T}^2) = 0$, these outcomes minimize to $f(\mathcal{R})$ theory. When we consider $(\mathcal{R} = \mathcal{R}_0)$ and $f_2(\mathbf{T}^2)$ as constants, then Equation (32) becomes

$$
\begin{aligned}
\frac{\mathbb{C}\sqrt{3}}{2} =\ & \left[\left\{\frac{1}{f_\mathcal{R}}\left(\text{æ} + \mathrm{P_r} - \mathrm{P}_\phi - 2\pi\mathbb{E}^2 + \frac{\mathcal{A}_0}{2}\right)\right\}^2 - \left(\text{æ} + 2\pi\mathbb{E}^2 + \frac{\mathcal{A}_0}{2}\right) \right. \\
& \times \frac{1}{f_\mathcal{R}}\left\{\left(\mathrm{P_r} - 3\mathrm{P}_\phi + 2\zeta\Theta - 4\pi\mathbb{E}^2 - \mathcal{A}_0\right) + \left(\mathrm{P_r} - \zeta\Theta - 2\pi\mathbb{E}^2 + \frac{\mathcal{A}_0}{2}\right)\right. \\
& \left.\left. \times \left(\mathrm{P}_\phi - \zeta\theta + 2\pi\mathbb{E}^2 - \frac{\mathcal{A}_0}{2}\right)\right\}\right]^{\frac{1}{2}},
\end{aligned}
\tag{33}
$$

where $\mathcal{A}_0$ is constant. This shows that principal stresses, electromagnetic field, and bulk viscosity induce inhomogeneity in the matter configuration. The relation of the Weyl tensor and inhomogeneity gives the gravitational arrow of time as indicated by the Penrose proposal [27]. This concept is based on the fact that tidal forces enhance the system's inhomogeneity during evolution. To obtain conformally flat spacetime, we consider the dust matter configuration as well as the zero charge that yields

$$\frac{\sqrt{3}}{2}\mathbb{C}' = \frac{1}{f_\mathcal{R}}\rho'. \tag{34}$$

This shows that $\mathbb{C} = 0 \iff \rho' = 0$, which implies that homogeneity in the energy density yields the conformally flat spacetime and vice versa.

## 6. Final Remarks

Recent cosmological observations, such as cosmic microwave background radiation, clustering spectrum, weak lensing, Planck data, supernovae type Ia, large scale structures, and galaxy redshift surveys, have revealed that the universe is expanding at an accelerated rate [49–52]. Modified theories of gravity have become a paradigm in the description of the gravitational interaction and its impact on cosmic expansion. Theories with extra gravitational fields, spatial dimensions, and higher derivatives are named as modified theories. A large number of approaches have been proposed to describe the cosmic acceleration based on the modification of Einstein's theory.

The $f(\mathcal{R}, \mathcal{T}^{\alpha\beta}\mathcal{T}_{\alpha\beta})$ theory, which is a generalization of $f(\mathcal{R})$ gravity and represents an alternative gravitation theory containing non-minimal curvature matter coupling, has gained much attention in recent years. Its gravitational action includes an additional force (a contraction of the energy-momentum tensor) together with the function of the Ricci scalar, which further modifies the gravitational interaction. This additional force is always appealing due to the reduction in the collapse rate. Hence, the addition of an extra term $(\mathcal{T}^{\alpha\beta}\mathcal{T}_{\alpha\beta})$ in a modified Einstein–Hilbert action provides a better description for unveiling the cosmic mysteries.

Gravitational collapse is considered a critical issue in general relativity and plays an important role in the structure formation of the universe. The study of gravitational collapse has become a subject of great interest for astrophysicists after the observational studies of gravitational waves using laser interferometric detectors, such as LIGO, TAMA, VIRGO, and GEO [53]. Almost all theoretical investigations into the processes of gravitational emissions have assumed that spacetime singularities are hidden inside black holes.

However, if the spacetime singularities are not enclosed by event horizons then the situation could be completely different from the case with horizons present. Such a spacetime singularity is called a globally naked singularity. Since there is no event horizon in the neighborhood of a globally naked singularity, Nakamura et al. [54] conjectured that a massive spacetime curvature could propagate away to infinity in the form of gravitational radiation. Therefore, the mass of the naked singularity would be lost through large gravitational emissions.

The cosmic censorship conjecture states that a spacetime singularity forms in the collapsing phenomenon that cannot be seen by the naked eye [55]. Regarding the formation of a naked singularity, spherically symmetric systems have been studied, as they are simple but possess rich physical content [56]. However, in some ways, such a system is too simple, as there is no gravitational radiation. Therefore, to study the general mechanism of gravitational radiation, one can add non-spherical perturbations to this system or can consider non-spherically symmetric spacetimes. In this regard, the cylindrically symmetric gravitational collapse has a significant physical meaning, because there is a degree of gravitational radiation, and further the spacetime singularity in this system is naked.

This paper investigated the dynamics of cylindrical collapse with charged dissipative fluid in the background of EMSG. We established the equations of motion with an electromagnetic field, which provided the correlation between energy and gravitational mass leading to the gravitational force. For the smooth matching of inner and outer spacetime, We considered Darmois junction conditions. We found that the difference between the masses of both spacetimes was $\frac{\mathbb{L}}{8}$, which is due to the unsatisfactory definition of the Throne C-energy. We applied the Misner–Sharp strategy to examine how the total energy changes with the time and radial coordinates. We, then, coupled dynamical and transport equations to investigate the effects of the correction terms and dynamical forces on the rate of collapse. The relation between the physical quantities and Weyl tensor was also established.

The proper time and radial derivatives of C-energy showed that the entire energy of the cylindrical star increases due to the effective radial stress and energy density, whereas the overall energy reduces because of the repulsive Coulomb effects. We found that the term $(Y)$ is directly related to the thermal conductivity and temperature and is inversely proportional

to the gravitational mass density, which depends on the effective fluid parameters and is independent of the electromagnetic field. The combined impact of the hydrodynamical forces and mass densities via the transport equation was also investigated. The entity $(Y)$ strongly affects these dynamical forces and describes the fate of the collapse rate. The various values of $(Y)$ and the dynamical equations yielded the following outcomes.

- If $Y = 1$ then dissipation takes place due to thermal conductivity as well as temperature. Furthermore, the influence of the hydrodynamical force minimizes the collapsing process due to the positive impact of the gravitational force.
- For $Y < 1$, the collapse rate reduces due to the anti-gravitational and Coulomb repulsive forces.
- When $Y > 1$, the gravitational mass density and attractive force enhance and increase the collapse rate.
- The overall systematic energy reduces because of the electromagnetic field, which implies that heat is dissipated in the outward direction.
- The additional modified terms yield repulsive effects and support the equilibrium state of the object and, hence, prevent the collapse rate.
- We analyzed that homogeneity in the energy density yielded conformally flat spacetime and vice-versa.

We found that the collapse rate was reduced for all values of $Y$ due to the Coulomb repulsive force, dissipative matter, and anti-gravitational behavior of the correction terms. Hence, the collapse rate was slower when compared with GR.

**Author Contributions:** M.S. suggested the research problem and finalized the manuscript while M.Z.G. did the calculations and prepared the initial draft. All authors have read and agreed to the published version of the manuscript.

**Funding:** No finding received.

**Conflicts of Interest:** There is no conflict of interest.

## Appendix A

The values of the correction terms are:

$$
\begin{aligned}
\mathcal{T}_{00}^{(EMSG)} &= \frac{\mathcal{W}^2}{2}(\mathcal{R}f_{\mathcal{R}} - f) + 2\mathcal{W}^2\left(h^2 - 2\rho^2 - \rho(\mathrm{P_r} + \mathrm{P}_\phi + \mathrm{P_z})\right)f_{\mathbf{T}^2} \\
&+ \frac{\mathcal{W}^2}{\mathcal{X}^2}\left(f_{\mathcal{R}}'' - \frac{\mathcal{X}\dot{\mathcal{X}}}{\mathcal{W}^2}\dot{f}_{\mathcal{R}} - \frac{\mathcal{X}'}{\mathcal{X}}f_{\mathcal{R}}'\right) + \frac{\mathcal{W}^2}{\mathcal{Y}^2}\left(\frac{\mathcal{Y}\mathcal{Y}'}{\mathcal{X}^2}f_{\mathcal{R}}' - \frac{\mathcal{Y}\dot{\mathcal{Y}}}{\mathcal{W}^2}\dot{f}_{\mathcal{R}}\right), \\
\mathcal{T}_{11}^{(EMSG)} &= \frac{\mathcal{X}^2}{2}(f - \mathcal{R}f_{\mathcal{R}}) + \mathcal{X}^2\left\{(\rho - \mathrm{P_r})(\mathrm{P}_\phi + \mathrm{P_z}) - \left(\rho^2 - \mathrm{P_r^2}\right) - 2h^2\right\}f_{\mathbf{T}^2} \\
&+ \frac{\mathcal{X}^2}{\mathcal{W}^2}\left(\ddot{f}_{\mathcal{R}} - \frac{\mathcal{W}\mathcal{W}'}{\mathcal{X}^2}f_{\mathcal{R}}' - \frac{\dot{\mathcal{W}}}{\mathcal{W}}\dot{f}_{\mathcal{R}}\right) - \frac{\mathcal{X}^2}{\mathcal{Y}^2}\left(\frac{\mathcal{Y}\mathcal{Y}'}{\mathcal{X}^2}f_{\mathcal{R}}' - \frac{\mathcal{Y}\dot{\mathcal{Y}}}{\mathcal{W}^2}\dot{f}_{\mathcal{R}}\right), \\
\mathcal{T}_{22}^{(EMSG)} &= \frac{\mathcal{Y}^2}{2}(f - \mathcal{R}f_{\mathcal{R}}) + \mathcal{Y}^2\left\{(\rho - \mathrm{P_{C\!E}})(\mathrm{P_r} + \mathrm{P_z}) - \left(\rho^2 - \mathrm{P_{C\!E}^2}\right)\right\}f_{\mathbf{T}^2} \\
&+ \frac{\mathcal{Y}^2}{\mathcal{W}^2}\left(\ddot{f}_{\mathcal{R}} - \frac{\mathcal{W}\mathcal{W}'}{\mathcal{X}^2}f_{\mathcal{R}}' - \frac{\dot{\mathcal{W}}}{\mathcal{W}}\dot{f}_{\mathcal{R}}\right) - \frac{\mathcal{Y}^2}{\mathcal{X}^2}\left(f_{\mathcal{R}}'' - \frac{\mathcal{X}\dot{\mathcal{X}}}{\mathcal{W}^2}\dot{f}_{\mathcal{R}} - \frac{\mathcal{X}'}{\mathcal{X}}f_{\mathcal{R}}'\right),
\end{aligned}
$$

$$
\begin{aligned}
\mathcal{T}_{33}^{(EMSG)} &= \frac{1}{2}(f - \mathcal{R}f_{\mathcal{R}}) + \left\{ (\rho - \mathrm{P}_z)(\mathrm{P}_\phi + \mathrm{P}_{\mathrm{r}}) - \left( \rho^2 - P_z^2 \right) \right\} f_{\mathbf{T}^2} \\
&\quad + \frac{1}{\mathcal{W}^2}\left( \ddot{f}_{\mathcal{R}} - \frac{\mathcal{W}\mathcal{W}'}{\mathcal{X}^2}f'_{\mathcal{R}} - \frac{\dot{\mathcal{W}}}{\mathcal{W}}\dot{f}_{\mathcal{R}} \right) - \frac{1}{\mathcal{X}^2}\left( f''_{\mathcal{R}} - \frac{\mathcal{X}\ddot{\mathcal{X}}}{\mathcal{W}^2}\dot{f}_{\mathcal{R}} - \frac{\mathcal{X}'}{\mathcal{X}}f'_{\mathcal{R}} \right) \\
&\quad + \frac{1}{\mathcal{Y}^2}\left( \frac{\mathcal{Y}\ddot{\mathcal{Y}}}{\mathcal{W}^2}\dot{f}_{\mathcal{R}} - \frac{\mathcal{Y}\mathcal{Y}'}{\mathcal{X}^2}f'_{\mathcal{R}} \right), \\
\mathcal{T}_{01}^{(EMSG)} &= (-3\rho + \mathrm{P}_{\mathrm{r}} - \mathrm{P}_\phi - \mathrm{P}_z)f_{\mathbf{T}^2} + \dot{f_{\mathcal{R}}}' - \frac{\mathcal{W}'}{\mathcal{W}}\dot{f}_{\mathcal{R}} - \frac{\dot{\mathcal{X}}}{\mathcal{X}}f'_{\mathcal{R}}.
\end{aligned}
$$

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
