# Peer review of "Dynamical Analysis of Charged Dissipative Cylindrical Collapse in Energy-Momentum Squared Gravity"

_universe, doi:10.3390/universe7050154_

Round 1
Reviewer 1 Report
I find this article very interesting. It is written clearly and correctly, is devoted to the study of important questions of cosmology and astrophysics, and contains a new approach to solving fundamental problems (in particular, to the primordial singularity problem). Therefore, I support the publication of this article in the journal Universe but with taking into account of two remarks:
- Since this work is inherently directly related to experiment (observational cosmology), then in my opinion a small part (several sentences) should be added to it discussing possible experiments for confirmation its main results.
- Minor remark. In reference [25] of this article (Bronnikov, K.A. and Kovalchuk, M.A....) the title of the journal is missing.
Reviewer 2 Report
I fail to see the interest in studying a highly artificial configuration (cylindrical) in an exotic theory of gravity with further exotic matter content. It just appears a calculation that is done because it can be done rather than an attempt to make any progress in physics. I cannot recommend publication.
Reviewer 3 Report
In this paper, the authors study the dynamics of charged cylindrical collapse with dissipative matter configuration in f (R, T_ab T^ab) theory which resolves the primordial singularity and provides feasible cosmological results. What makes significant the manuscript is the discussion on the modified terms, dissipative matter and electromagnetic field which reduce the collapsing phenomenon. Therefore, I recommend the paper for publication in Universe.
Author Response
The modified terms make the manuscript more significant because the
correction terms also affect the matter variables whereas charged
yields the same behavior as in other gravitational theories.
However, the higher-order curvature terms as well as squared
entities of fluid parameters make the system more stable and reduce
the collapse rate as compared to general relativity and other
alternative gravitational theories.
Round 2
Reviewer 2 Report
I stand by my original report. Models with cylindrical symmetry teach us little about what really happens in nature, particularly in present times when detailed numerical simulations of realistic gravitational collapse exist. This paper should be rejected.